

# Automatic detection of teacher behavior in classroom videos using AlphaPose and Faster R-CNN algorithms

Jing Huang[1,2], Harwati Hashim[1], Helmi Norman[1], Mohammad Hafiz Zaini[1] and Xiaojun Zhang[2]

[1] Faculty of Education, Universiti Kebangsaan Malaysia, Selangor, Malaysia
[2] Faculty of Economics and Business Foreign Languages, Wuhan Technology and Business University, Wuhan, Hubei, China

## ABSTRACT

This study proposes an automated classification framework for evaluating teacher behavior in classroom settings by integrating AlphaPose and Faster region-based convolutional neural networks (R-CNN) algorithms. The method begins by applying AlphaPose to classroom video footage to extract detailed skeletal pose information of both teachers and students across individual frames. These pose-based features are subsequently processed by a Faster R-CNN model, which classifies teacher behavior into appropriate or inappropriate categories. The approach is validated on the Classroom Behavior (PCB) dataset, comprising 74 video clips and 51,800 annotated frames. Experimental results indicate that the proposed system achieves an accuracy of 74.89% in identifying inappropriate behaviors while also reducing manual behavior logging time by 47% and contributing to a 63% decrease in such behaviors. The findings highlight the potential of computer vision techniques for scalable, objective, and real-time classroom behavior analysis, offering a viable tool for enhancing educational quality and teacher performance monitoring.

Corresponding author
Xiaojun Zhang, whgs2014@163.com

# INTRODUCTION

The theory and practice of basic education have matured considerably, allowing virtually everyone to receive a comprehensive basic education. Although significant progress has been made in the field of education and teaching, accumulating a wealth of teaching experience, there are still problems and shortcomings in some aspects (*Yoshikawa et al., 2013*). For instance, in higher education, the personal level of the teacher significantly affects the overall quality of education quantitatively. However, since undergraduates lack autonomous judgment, it is difficult to evaluate the behavior of teachers quantitatively.

Instructional behavior analysis aims to analyze the behaviors of both teachers and students during the teaching process to explore the underlying connections and patterns in behavior occurrence and development (*Kirkpatrick, Akers & Rivera, 2019*). Traditional classroom observation methods mainly rely on recording teacher-student behaviors and classroom activities by education personnel, which cannot be used sustainably. In contrast, while the questionnaire method has good reliability and validity, it is subjectively

influenced by individual opinions. Therefore, utilizing objective data and adopting efficient analytical methods is an inevitable requirement for the development of instructional behavior analysis.

Objective data is not influenced by individual subjectivity and can more accurately reflect the actual teaching and learning status of individuals. Recently, due to the diversification of sensing device forms and performance development, wearable sensors have been widely applied in the field of education (*Shaw et al., 2019*). However, this method is subject to device limitations. Efficient behavioral analysis methods should meet two conditions. In terms of data collection, it should have low hardware requirements, not cause too much interference to teaching, and be able to be used in a sustainable manner. In terms of data analysis, automation, speed, accuracy, diversity, and intelligence are the trends of development (*Li, Ortegas & White, 2023*).

Classroom videos, which contain both image and sound information, are a common way to record behavioral data and can accurately recreate behavior details during teaching without causing interference to the classroom (*Li et al., 2025*). Although classroom videos contain a lot of noise and are subject to technical limitations, they have been mainly used as tools for classroom reproductions rather than for mining their visual information (*Jung, Kim & Jain, 2004*; *Wang et al., 2023*; *Hu et al., 2022*). However, with the in-depth development of deep learning technology in the field of computer vision, neural network algorithms, represented by convolutional neural networks, have become the mainstream method for image recognition (*Xie et al., 2021*). Compared with traditional manually designed feature algorithms, neural network algorithms have stronger feature extraction and expression capabilities, greatly improving the accuracy of image recognition and providing strong technical support for automating the recognition and analysis of instructional behaviors (*Albawi, Mohammed & Al-Zawi, 2017*). The intellectualization of classroom instructional behavior analysis can greatly reduce the time and effort that teachers spend on teaching analysis and has significant practical significance for promoting and innovating teaching analysis methods.

The human visual system is influenced by the environment, physiological deficiencies, and mental rules, which make it impossible for observers to comprehensively and accurately perceive changes in a student's classroom state. Based on the principles of human visual perception, computer vision can achieve similar functions to those of the eyes and brain, which enables its effective application in the education industry (*Poudyal, Mohammadi-Aragh & Ball, 2022*), such as using computer vision technology to identify educational environments and applying it to recognize the behaviors of teachers.

The emergence of video acquisition equipment and the popularization of information classrooms with cameras have produced a large number of classroom teaching videos (*Akçayır & Akçayır, 2018*). There are several advantages of instructional videos. Firstly, they record comprehensive data, which is suitable for observing student behavior in the classroom. Secondly, the observer's temporal and spatial dimensions of observation become more flexible and extensive. Thirdly, the video-recorded data is objective and can accurately reproduce the teaching scenario. As observing student behavior can provide effective information for objective classroom teaching evaluations, the effective utilization

of classroom videos is a method for improving classroom teaching evaluations. The development of computer vision technology has provided the possibility for realizing video-based observation of student classroom behaviors (*Ngoc Anh et al., 2019*).

Given these advancements, applying computer vision techniques to automatically monitor and evaluate teacher behaviors presents a promising research direction, especially in classrooms where traditional observation methods face limitations in objectivity and scalability. Although various deep learning-based behavior detection methods have been developed, challenges remain in accurately detecting inappropriate teacher behavior due to issues with model interpretability and high computational requirements. Therefore, this study proposes a method that integrates AlphaPose and Faster region-based convolutional neural network (R-CNN) to enhance the precision and applicability of classroom behavior detection.

The main contributions of this article are summarized as follows:

A system for analyzing the behavior of teachers has been designed and implemented, which is capable of identifying inappropriate behavior exhibited by said teachers.

A classification model combining AlphaPose and Faster R-CNN algorithms was established based on the PCB dataset, and the effectiveness of the classification model was evaluated using the F2-score evaluation standard.

Analyzed and discussed the potential application of automatic behavior classification methods for teachers in higher education and concluded that emerging technologies could assist educational work and are the development direction of future smart classrooms.

## RELATED WORKS

The purpose of video behavior classification is to classify video behaviors into predefined categories based on video content (*Awad & Motai, 2008*). Images themselves contain a wealth of information, and videos are an extension of images in the temporal dimension, often containing around 25 frames per second and occupying multiple times the storage space compared to images. Storing and analyzing this video content requires enormous financial and human resources. Before the widespread application of computer-automated video data analysis, the classification of video content behavior was generally manually performed, with low efficiency and high rates of misjudgment and omission (*Pisarev et al., 2019*). The promotion of automatic video content analysis technology has broad and far-reaching practical significance. Computer video behavior analysis technology can not only automatically monitor multiple signals at the same time without generating fatigue, but also reduce the possibility of misjudgment. Its application in the field of video content retrieval can significantly reduce the workload of public safety personnel performing video retrieval, improve their retrieval efficiency, and reduce omission rates. The promotion of automatic video content analysis technology has broad and far-reaching practical significance.

Object detection is a hot research area in computer vision technology and has wide application in real life, with significant practical significance. Since 2012, the rapid development and widespread application of neural networks have promoted the rapid growth of object detection (*Xu, Park & Zhang, 2020*). In 2014, *Girshick, Donahue &*

*Darrell (2014)* proposed the R-CNN object detection algorithm. R-CNN combines AlexNet with selective search algorithms, which significantly improves detection accuracy but with low performance. Studies have shown that the convolution layer of convolutional neural networks has a good ability to locate target positions, but fully connected layers weaken this ability. In 2015, *Ren et al. (2015)* proposed the Faster R-CNN detection algorithm. Faster R-CNN achieved good results in terms of detection speed and classification accuracy.

Classroom teaching analysis helps to improve the quality of teaching. With the rapid development of artificial intelligence technology, there have been significant changes in the forms of teaching analysis and evaluation. Compared with traditional manual observation and recording methods, AI technology has greater advantages in providing continuous feedback and process evaluation. This is because the technology not only automates the laborious work of recording, storing, and analyzing data, saving educators time and energy, but also brings more advanced methods for information acquisition. In recent years, research on AI education applications involving different teaching environments, data formats, application methods, and design patterns has shown an increasing trend year by year. *Shaw et al. (2019)* collected sequential information such as student activities, phone status, and dialogue through a StudentLife mobile app while combining covariates such as sleep duration or test periods to personalize predictions of students' stress states using deep multitask networks. *Yu, Wu & Liu (2019)* utilized online Massive Open Online Courses (MOOCs) course video clickstream data to predict whether students can obtain good learning outcomes by training k-nearest neighbors algorithm (KNN), support vector machines (SVM), and artificial neural network (ANN) models with certificate acquisition as labels, resulting in an accuracy rate above 95%, which will help feedback to teachers for improving teaching effectiveness thereby enhancing MOOC completion rates.

Recent advancements in object detection have yielded models such as YOLOv8 and Detection Transformer (DETR). YOLOv8 (You Only Look Once version 8) emphasizes real-time detection by optimizing both inference speed and accuracy, making it highly suitable for edge computing environments. On the other hand, DETR introduces a transformer-based architecture that eliminates the need for region proposal networks, enabling end-to-end object detection with high precision. Despite their merits, these models either lack the integration of pose-based semantic features or are less interpretable in educational applications.

Overall, the academic community has conducted in-depth research on computer vision-based human pose recognition algorithms and formed a relatively mature theoretical framework through long-term exploration. However, most of the aforementioned studies only use object detection algorithms to analyze static images or videos. Regarding the characteristics of multiple targets in teaching scenarios, we believe that identifying and locating human limbs first and then analyzing their behaviors by combining limb position information would yield better results.

## METHODS

This article proposes a Faster R-CNN classification method based on local posture for teaching videos in classrooms, which detects the behaviors of teachers in class.

### Process introduction

The experimental process is shown in Fig. 1.

AlphaPose is an efficient open-source human pose estimation system capable of detecting and tracking multiple persons' body joints in real-time. In this study, each frame in the PCB dataset is processed through AlphaPose to extract 17 key skeletal points per person, including joint coordinates of the head, shoulders, elbows, wrists, hips, knees, and ankles. These key points are then converted into feature vectors representing spatial relationships of limb positions. To ensure scale invariance and consistency across frames, all coordinate values are normalized by image width and height.

Prior to feeding the data into the Faster R-CNN classifier, we perform data augmentation techniques, including horizontal flipping and Gaussian noise injection, to enhance model robustness. The processed keypoint vectors form a high-dimensional feature representation that encapsulates the teacher's motion and posture information, which is then used by the classifier to identify inappropriate behavior patterns. This method can be used for the analysis of classroom videos and the detection of teacher behavior. Compared to single behavioral classification methods, combining AlphaPose with Faster R-CNN classification methods results in higher recognition accuracy. The process of behavioral classification is shown in Fig. 2.

### Dataset introduction

We used a PCB dataset to train a model that classifies the behavior of teachers. The dataset includes 74 video clips with a total of 51,800 labeled frames after processing. The data set is available at https://doi.org/10.1007/s00521-020-05587-y. Each frame contains multiple preschoolers and one teacher and is labeled either 0 (no inappropriate behavior by the teacher towards the children) or 1 (inappropriate behavior by the teacher towards the children). A total of 2,937 marked images were identified as containing improper actions, which represents 5.67% of the overall number of frames. To optimize our data usage, we partitioned the dataset into training and test sets, maintaining a ratio of 7:3. The distribution of data types after partitioning is presented in Table 1.

To improve the generalization ability of the model and simulate the diversity of real-world classroom environments, a series of data augmentation techniques were applied to the training set. These include:

- **Random horizontal flipping:** to account for mirrored teacher gestures and body orientations.
- **Random rotation (±15 degrees):** to simulate camera angle variations and minor shifts during video capture.
- **Random scaling and cropping:** to emulate varying distances between the camera and subjects, as well as partial occlusions.

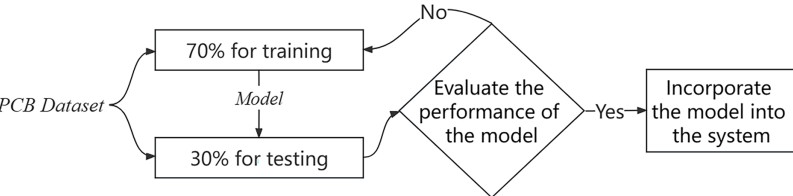

**Figure 1 Experimental process.**

- **Brightness and contrast adjustments:** to reflect lighting changes that may occur across different classroom conditions.

These augmentation strategies were implemented using standard OpenCV and PyTorch-based data pipelines before feature extraction by the AlphaPose algorithm.

## Experimental method

After the preparation work on the dataset is completed, we use the AlphaPose model to recognize the body limbs of people in the video dataset. AlphaPose is an efficient open-source human pose estimation system that can achieve multiple-person pose estimation, pose tracking and other functions.

The video frames recognized by AlphaPose will be labeled with the body information of the children and teachers. Compared to the original video, data that has been labeled is easier to classify correctly.

In order to avoid the loss function failing to converge during training and enhance the generalization of the model, it is necessary to normalize the sample data (*Liu et al., 2025*; *Wang et al., 2022*). This article mainly performs normalization processing on the skeleton point coordinates extracted by the AlphaPose model, as shown in Formula (1).

$$(\dot{x}, \dot{y}) = \left(\frac{x}{w}, \frac{y}{h}\right). \tag{1}$$

Among them, $x$ and $y$ are the $x$ coordinate and $y$ coordinate of the skeletal points in the image, respectively. $w$ denotes the width of the image, and $h$ denotes the height of the image. Ultimately, the normalized coordinates $(\dot{x}, \dot{y})$ are obtained.

We divided the training set and test set in a ratio of 7:3 during the training process. We chose the Sigmoid function as the activation function, which is represented as Formula (2). In this article, recognizing teacher behavior is a binary classification problem, and we selected the binary cross-entropy loss function as our loss function. Its specific calculation method is shown in Formula (3).

$$sigmoid(y) = \frac{1}{1 + e^{-y}} \tag{2}$$

$$L(y, \hat{y}) = -\frac{1}{2}[\hat{y}\log(sigmoid(y))] + (1 - \hat{y})\log(sigmoid(1 - y)). \tag{3}$$

Among them, $y$ represents the actual neuron, and $\hat{y}$ represents the true label marked in the dataset. 0 indicates normal behavior, while 1 indicates abnormal behavior.

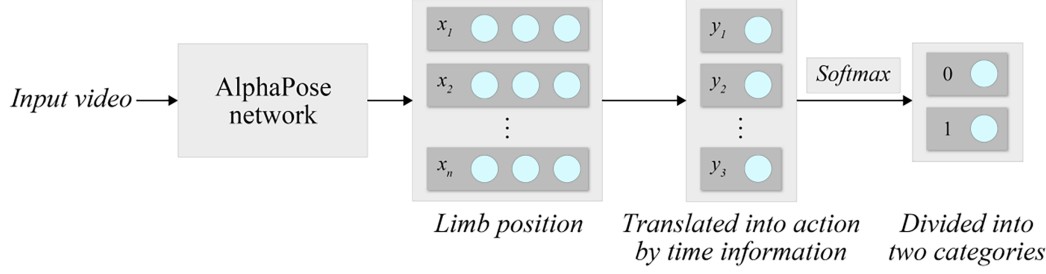

**Figure 2 System processing.**

**Table 1 The number of different categories of data.**

| Category | Training | Testing | Total |
| --- | --- | --- | --- |
| 0 | 34,204 | 14,659 | 48,863 |
| 1 | 2,056 | 881 | 2,937 |
| Total | 36,260 | 15,540 | 51,800 |

Compared to other convolutional neural network models, Faster R-CNN performs well in terms of both detection speed and classification accuracy. Therefore, this article uses this network to classify images processed by AlphaPose.

Faster R-CNN, a deep learning-based object detection algorithm by Ross Girshick in 2015 and updated in 2016, represents an improved version of R-CNN and Fast R-CNN. It introduces the region proposal network (RPN) to replace the Selective Search algorithm for object extraction, thereby enhancing both detection speed and accuracy.

The Faster R-CNN model consists of two principal components: the region extraction network and the object classification network. The RPN, a sub-network based on convolutional neural networks, generates candidate object regions in the image. RPN slides a small window on the feature map, and each window outputs the scores of k (usually 9) anchor boxes and their corresponding bounding box regression parameters. These anchor boxes have different ratios and scales, covering objects of varying sizes and shapes. Then, based on the scores of these anchor boxes, the non-maximum suppression (NMS) algorithm selects the most likely candidate regions containing the object, which are then sent to the object classification network. The object classification network classifies and locates the regions. It receives the candidate regions extracted from the RPN and classifies and locates each region. This network usually uses a fully convolutional network, which can process regions of different sizes and output which category each region belongs to and its corresponding bounding box position.

The Faster R-CNN training process consists of two steps: first, the RPN network generates candidate regions and classifies and locates them. Then, the network is trained using the cross-entropy loss function and the smooth L1 loss function to correctly classify and locate the object regions, as shown in Formulas (4) and (5).

$$Loss = -\sum_{i=0}^{n} x_i log(\hat{x}_i) \tag{4}$$

$$Smooth\ L_1 = \begin{cases} |x| - 0.5 & |x| > 1 \\ 0.5x^2 & |x| < 1. \end{cases} \tag{5}$$

Relative to Fast R-CNN, Faster R-CNN has made significant strides in detection speed and accuracy. Its speed is more than ten times faster than Fast R-CNN, and its accuracy has also improved. Innovative in region extraction and object classification, Faster R-CNN stands as one of the most advanced object detection algorithms to date.

### Evaluation method

After completing the training, we can obtain a Faster R-CNN model for the behavior classification of teachers. For each image, the output of the classification model consists of two classes: 0 represents normal teacher behavior and 1 represents inappropriate teacher behavior. The classification results include four categories: true positive (TP) indicates that positive samples are correctly classified as positive by the model; false positive (FP) indicates that negative samples are incorrectly classified as positive by the model; false negative (FN) indicates that positive samples are incorrectly classified as negative by the model; and true negative (TN) indicates that negative samples are correctly classified as negative by the model. Evaluating algorithm performance requires considering both precision (P) and recall (R), with formulas for calculating precision and recall given in Formulas (6) and (7), respectively (*Torgo & Ribeiro, 2009*). Since recognition models in this scenario tend to favor recall rate, we selected F2-score, a commonly used comprehensive evaluation index for classification models, as our evaluation standard using Formula (8).

$$Precision = \frac{TP}{TP + FP} \tag{6}$$

$$Recall = \frac{TP}{TP + FN} \tag{7}$$

$$F_2\text{-}score = \frac{5 * Recall * Precision}{4 * Recall + Precision}. \tag{8}$$

## EXPERIMENT

### Dataset preview

Using the method introduced in the previous section, this article uses a PCB dataset. First, the AlphaPose algorithm is used to recognize actions in all labeled images in the dataset. The recognition results of the dataset include limb information of undergraduates and teachers in the images. The position information of human limbs is essentially the position information of limb key nodes, which can be regarded as multidimensional vectors. To enhance the visual representation of the data, we applied the t-SNE algorithm to project the data onto a two-dimensional plane for display, as depicted in Fig. 3.

As evident from the graph, a majority of the data mapping points are situated in the central region of the chart, whereas a negligible proportion of the data points are dispersed near the central cluster, signifying anomalous characteristics. The aberrant features of these data points are indicative of teacher misconduct. The t-SNE projection reveals a distinguishable clustering pattern, where outlier points corresponding to inappropriate

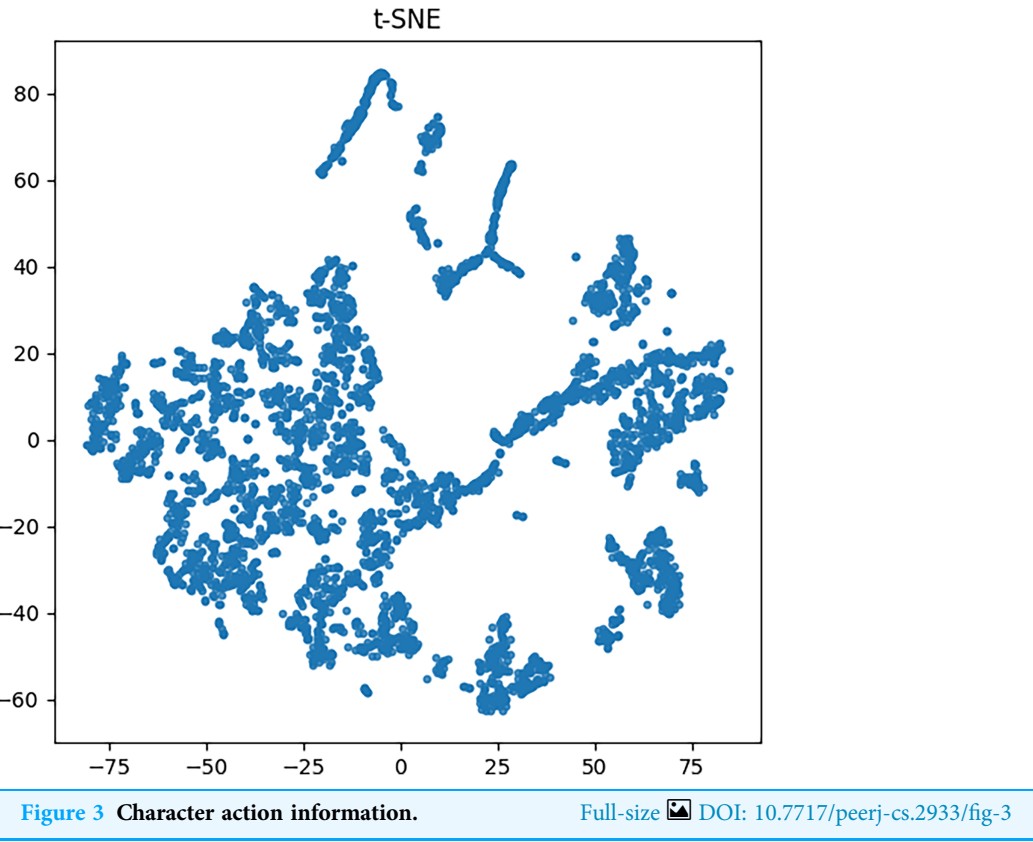

**Figure 3 Character action information.** Full-size 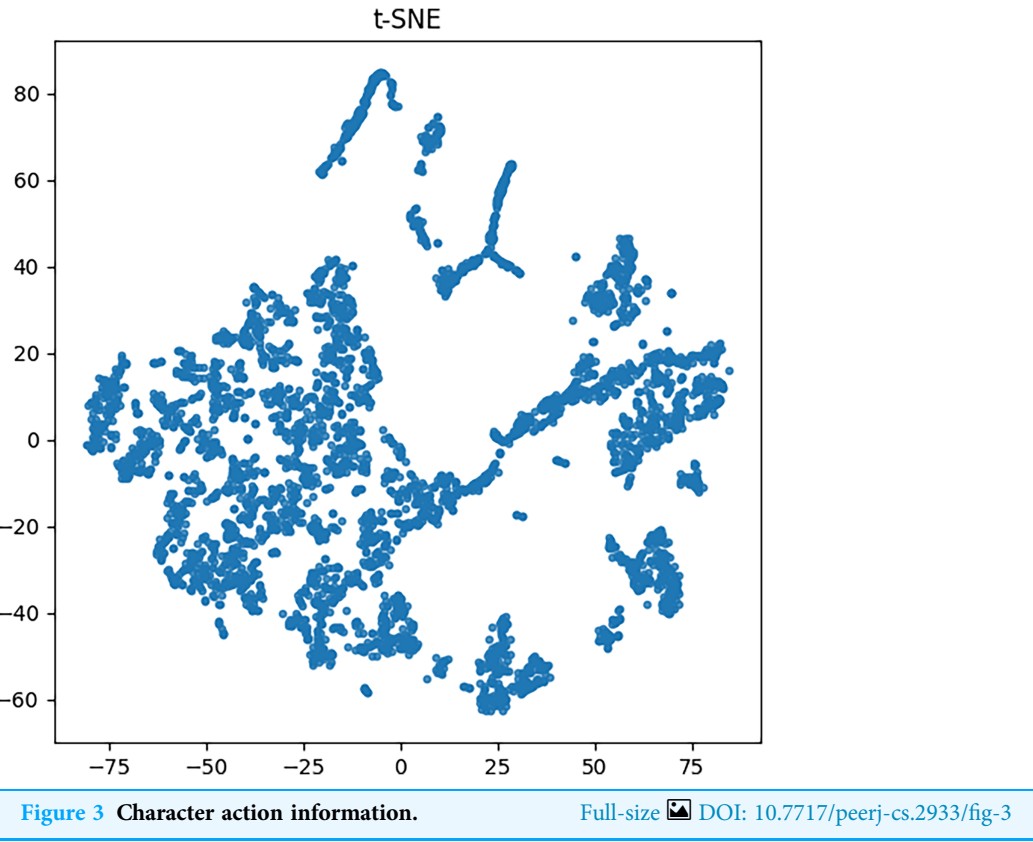 DOI: 10.7717/peerj-cs.2933/fig-3

teacher behaviors are separable from the central cluster. This demonstrates the discriminative capability of AlphaPose features.

## Model training

During the training phase, 70% of the available data was utilized as the training set. The training set consisted of 34,204 frames representing proper behavior and 2,056 frames displaying improper behavior, resulting in a total of 36,260 frames. Figure 4 illustrates the change in loss function as training progressed.

The figure depicts a gradual decrease in the loss function value as the number of training rounds increases. Initially, the loss function value rapidly diminishes with the increase of training rounds, followed by a conspicuous jolt at around 14 rounds. Subsequently, the value continues to decrease quickly. However, at approximately 35 rounds, the loss function value exhibits a minor amplitude oscillation, and it no longer decreases below 0.3, signifying that the model's performance has reached a bottleneck. After around 70 rounds, the model was optimized and escaped from the local optimal solution. The loss function value no longer reaches 0.4 and remains minimal. As training rounds progress, the loss function value continues to gradually decline. The loss function curve shows stable convergence after approximately 100 epochs, indicating that the model achieves a satisfactory trade-off between training accuracy and generalization.

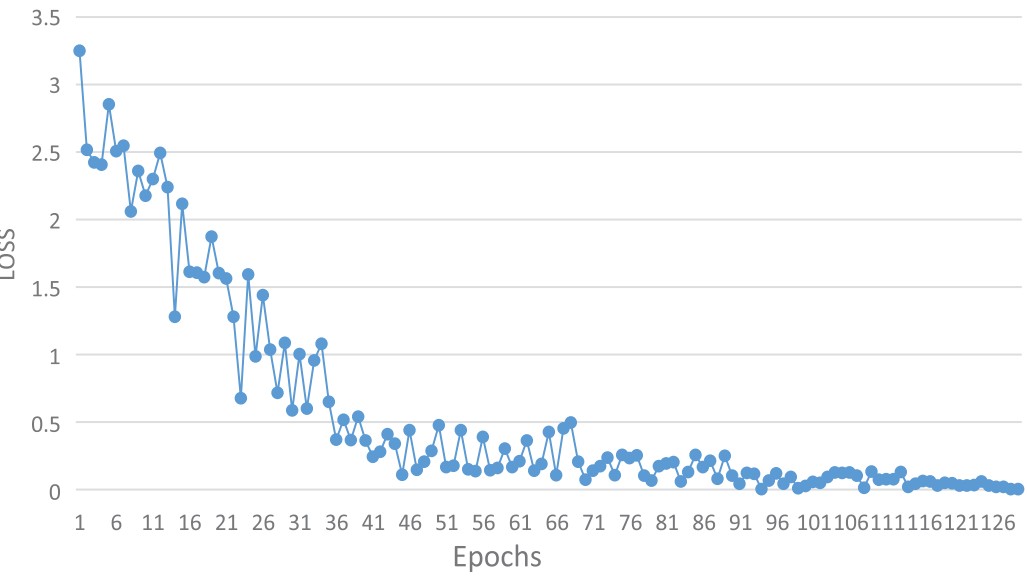

**Figure 4** **The change of loss function with epoch in training process.**

## Evaluation and results

After training is complete, the model is tested using test set data and the results are shown in the confusion matrix in Fig. 5.

The problem addressed in this article is a binary classification problem. Thus, the confusion matrix is a 2 * 2 matrix. As can be seen from the figure, TN is 42831 and TP is 6672. The precision rate is 0.7489, and the recall rate is 0.9911.

In this particular scenario, the recognition model prioritizes recall rate, and the evaluation standard for the model in this article is the F2-score. Table 1 presents a comparison between the impact of the Faster R-CNN classification method and other models following AlphaPose processing.

To quantify the contribution of AlphaPose in our framework, we interpret the Faster R-CNN model in Table 2 as a baseline without AlphaPose preprocessing. Compared to this configuration, the full system incorporating AlphaPose achieved a 1.03% improvement in F2-score and a 1.04% improvement in recall. This clearly demonstrates that the pose-based features extracted by AlphaPose significantly enhance classification accuracy by encoding motion and spatial cues that are otherwise difficult to capture using raw image features alone. The results presented in the table unequivocally demonstrate the superiority of the method posited in this article over other models. The proposed method boasts an accuracy rate that surpasses that of the basic Faster R-CNN model by 1.3% and a recall rate that surpasses the same model by 1.06%. In comparison to traditional CNN and SVM, the proposed method achieves an improvement in recall rate by 10.1% and 7.4%, respectively. Furthermore, the F2-score of the proposed method outperforms all other models, with a score that surpasses that of CNN, SVM and Faster R-CNN by 6.09%, 3.67% and 1.19%, respectively. This improvement highlights the advantage of integrating pose estimation

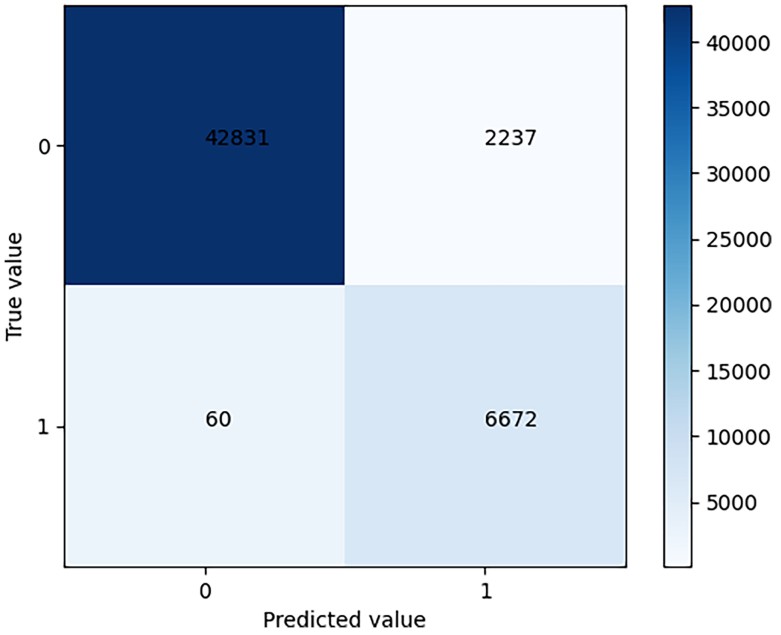

**Figure 5 Confusion matrix of test results.**     

with object detection for behavioral classification. The combination of AlphaPose and Faster R-CNN significantly enhances the detection accuracy for inappropriate teacher behaviors, which is critical for practical deployment in educational monitoring systems.

In a practical application in classrooms, the accuracy rate of the AlphaPose + Faster R-CNN classification method can reach 74.89%. As shown in the Fig. 6, after using this method, education staff reduced their time for recording and analyzing each child's behavior by 47%. There was a reduction of 63% in incidents of inappropriate behavior by teachers.

## DISCUSSION

Behavior analysis of teachers can help them understand the essence of the classroom and discover classroom patterns, thereby improving teaching quality and enhancing students' learning outcomes (*Powell et al., 2008*). This article aims to explore the current situation of improper behavior by teachers and the prospects for applying this method in classrooms. Student classroom behavior detection can be used to evaluate student performance and teacher quality, laying a foundation for intelligent evaluation in education (*Yu et al., 2017*).

Traditional quantitative analysis of teaching behavior is inefficient due to manual recording methods. With the continuous development of technology, models based on machine learning and computer vision for human pose recognition and prediction have been widely applied (*Le & Nguyen, 2013*), leading to increasing attention on automated analysis methods for classroom behavior. Among them, intelligence analysis based on informationized classroom video has good development prospects because it meets the needs of real classrooms with its low cost, non-interference, automation, and normalization advantages (*Yang et al., 2020*; *Wang et al., 2025*).

**Table 2 Comparison of AlphaPose+Faster R-CNN and other model.**

| Model | Precision | Recall | F2-score |
|---|---|---|---|
| CNN | 0.7975 | 0.9001 | 0.8775 |
| SVM | 0.8094 | 0.9230 | 0.8978 |
| Faster R-CNN | 0.7392 | 0.9807 | 0.9206 |
| AlphaPose+Faster R-CNN | 0.7489 | 0.9911 | 0.9309 |

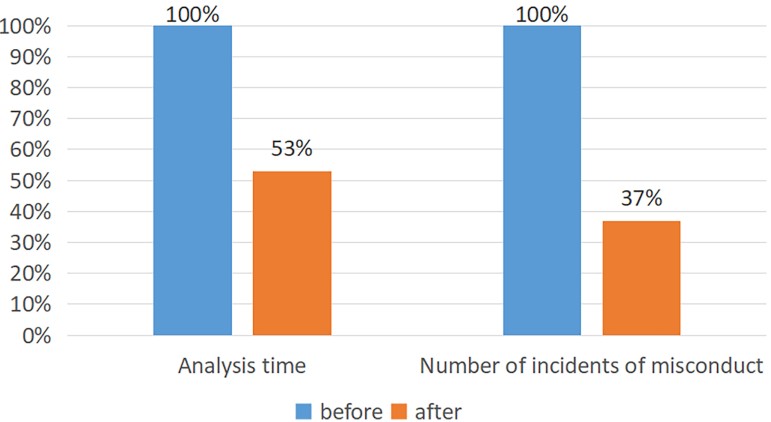

**Figure 6 The performance of the model after application.**

Advanced algorithms and computing devices now allow informationized classrooms to achieve similar effects using lower-cost software that eliminates hardware limitations (*Desmarais et al., 2021*). Moreover, frequent events involving excessive behavior by younger people in society have led people to pay more attention to standardizing higher practices (*Senthamilarasi et al., 2019*; *Ding et al., 2023*). Therefore, in the current situation of popularization of information technology courses, the application prospects of the methods proposed in this article are very broad. The following are several application prospects:

Improving teaching quality: Through the use of video monitoring technology and algorithms, teachers' teaching behaviors can be analyzed to identify and improve deficiencies in a timely manner, thereby improving teaching quality and efficiency.

Enhancing teaching effectiveness: By analyzing students' behaviors, their learning status and interests can be understood, allowing for targeted development of teaching strategies to improve student learning outcomes.

Promptly identifying problems: Monitoring both teachers and undergraduates' behavior enables prompt identification of teacher misconduct, bullying incidents involving undergraduates, or by undergraduates towards others.

Improving supervision and management efficiency: The use of intelligent algorithms significantly improves the effectiveness and accuracy of supervision while enabling 24-h monitoring of education classrooms. This not only enhances real-time precision and

scientificity in education management but also provides greater security assurance for students and parents.

Although there are privacy protection issues that need to be addressed, with continuous technological development, improvements in privacy protection measures will make it possible for this method to have broader application prospects. This method, which analyzes teacher behavior based on monitored video images, will provide favorable technical means supporting optimization in higher education while improving educational quality and effectiveness. In addition, although the dataset used in this study consists of 74 classroom videos from a specific preschool context, the proposed framework is designed with generalizability in mind. The modular structure of AlphaPose and Faster R-CNN allows adaptation to different educational settings, such as secondary schools or higher education classrooms. Future work will focus on cross-domain validation using datasets from diverse countries, subject areas, and classroom layouts to further verify the robustness and scalability of the system.

## CONCLUSION

This article designs and implements a teacher behavior recognition method that combines AlphaPose and Faster R-CNN algorithms to detect teachers' classroom behaviors.

### Experiment summary

Firstly, the PCB dataset is used as training data in this article. Then, the AlphaPose algorithm is used to identify the position of teachers' limbs, and the video with limb position information is input into the Faster R-CNN classification network for behavior classification. Experimental results show that the AlphaPose+Faster R-CNN model performs better than several other models. In the test set, TN is 42831, TP is 6672, the precision rate is 0.7489 and the recall rate is 0.9911. In actual classroom application, this system's accuracy can reach 74.89%, reducing educators' time spent on recording and analyzing each teacher's behavior by 47% while decreasing inappropriate behaviors by teachers by 63%. Finally, this article analyzes and discusses the applications and prospects of Faster R-CNN classification methods based on local posture in teacher behavior that is under increasing attention.

### Contribution

Moreover, this study underscores the immense potential of utilizing cutting-edge computer vision and machine learning techniques in the sphere of education. As technology continues to advance, there is an escalating necessity to incorporate these techniques into educational settings to augment teaching and learning outcomes. For instance, computer vision algorithms can be employed to monitor student engagement and attention levels during classroom activities, which can enable teachers to modify their teaching strategies and provide individualized feedback to students. Additionally, machine learning models can be utilized to scrutinize student performance data and offer personalized learning recommendations based on their unique strengths and weaknesses.

## Prospect

Another promising avenue of research in the realm of computers and education is the creation of intelligent tutoring systems. These systems can use artificial intelligence algorithms to deliver personalized learning experiences for students based on their specific learning styles and requirements. By analyzing student data in real time, these systems can adapt their instructional techniques and offer targeted feedback to facilitate students in accomplishing their learning objectives.

In summary, while this study constitutes a substantial advancement in the field of computers and education, there is still a considerable amount of work to be done to realize the full potential of technology in enhancing teaching and learning outcomes. Future research can concentrate on developing more sophisticated models and analysis frameworks, incorporating multimodal data, and exploring new applications of computer vision and machine learning in educational settings. Ultimately, the objective is to generate more effective and personalized learning experiences for students, thereby enhancing overall educational outcomes.

## Deficiency

It is worth noting that the dataset used in this article is limited in scope and sample size. Future studies may consider using larger datasets to verify the results of this study. In addition, to further improve behavior recognition accuracy, future research can more deeply optimize the AlphaPose algorithm and Faster R-CNN classification network to achieve better classification performance in broader contexts.

### Funding

This study is funded by Faculty of Education, Universiti Kebangsaan Malaysia under research grant No. GG-2024-030; this study is also funded by The Advantaged Characteristic Discipline Groups of Colleges and Universities of Hubei Province "Digital Commerce and Management". The funders had no role in study design, data collection and analysis, decision to publish, or preparation of the manuscript.

### Grant Disclosures

The following grant information was disclosed by the authors:
Faculty of Education, Universiti Kebangsaan Malaysia: GG-2024-030.
The Advantaged Characteristic Discipline Groups of Colleges.
Universities of Hubei Province "Digital Commerce and Management".

### Competing Interests

The authors declare that they have no competing interests.

### Author Contributions

- Jing Huang conceived and designed the experiments, performed the computation work, prepared figures and/or tables, and approved the final draft.

- Harwati Hashim performed the experiments, analyzed the data, prepared figures and/or tables, and approved the final draft.
- Helmi Norman conceived and designed the experiments, analyzed the data, prepared figures and/or tables, and approved the final draft.
- Mohammad Hafiz Zaini performed the experiments, performed the computation work, authored or reviewed drafts of the article, and approved the final draft.
- Xiaojun Zhang analyzed the data, authored or reviewed drafts of the article, and approved the final draft.

## Data Availability

The dataset is available at Kaggle: https://www.kaggle.com/datasets/kaiyueyyds/dataset-of-student-classroom-behavior.

The code is in the Supplemental File.

## Supplemental Information

Supplemental information for this article can be found online at http://dx.doi.org/10.7717/peerj-cs.2933#supplemental-information.

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
