# Peer review of "Automatic detection of teacher behavior in classroom videos using AlphaPose and Faster R-CNN algorithms"

_PeerJ Computer Science, doi:10.7717/peerj-cs.2933_

## Round 0.1 · original submission · Major Revisions

Dear authors

Thank you for your submission. Your manuscript has been reviewed by the experts in the field and you will see that they have a couple of good and constructive sugeestions for the improvement of your article. I agree with them so please revise and update your manuscript carefully in light of the comments of the experts and mine below. Please submit a detailed response to the comments.

Academic Editor Comments:

The current study uses the Classroom Behavior (PCB) dataset, which consists of 74 video clips. This sample size may not be large enough to capture the full variety of teaching environments and behaviors. Expanding the dataset to include more diverse classrooms (e.g., different age groups, subjects, or teaching methods) would increase the generalizability of the model's predictions.

The integration of AlphaPose and Faster R-CNN could be optimized further
it would be helpful to test the model's real-time performance in dynamic classroom environments.

The language must be improved.

**Language Note:** The Academic Editor has identified that the English language must be improved. PeerJ can provide language editing services - please contact us at [email protected] for pricing (be sure to provide your manuscript number and title). Alternatively, you should make your own arrangements to improve the language quality and provide details in your response letter. – PeerJ Staff

Reviewer 1 ·

Basic reporting

The introduction presents the research problem in a general manner but does not explicitly address the key challenges in behavior detection, such as the limitations of current methods regarding real-time performance, accuracy, and interpretability. To strengthen this section, it is recommended to include a concise paragraph at the end of the introduction that clearly defines the core research problem. For instance:

“Although various deep learning-based behavior detection methods have been developed, challenges remain in accurately detecting inappropriate teacher behavior due to issues with model interpretability and high computational requirements. Therefore, this study proposes a method that integrates AlphaPose and Faster R-CNN to enhance the precision and applicability of classroom behavior detection.”

In Section 3.1 (Methodology Introduction), it would be beneficial to include a more detailed explanation of how AlphaPose extracts skeleton keypoints and how this data is processed as input for Faster R-CNN. For instance:

“How does AlphaPose output keypoint information that can be utilized as features for Faster R-CNN? Are there any additional preprocessing steps, such as keypoint coordinate normalization or data augmentation?”

While Faster R-CNN remains a widely adopted model for object detection, recent advancements such as YOLOv8 (prioritizing real-time efficiency) and DETR (leveraging transformers for end-to-end detection) have demonstrated notable improvements in speed and accuracy. To provide a comprehensive literature review, the Related Works section should include a discussion of these alternatives, followed by a clear rationale for selecting Faster R-CNN.

Experimental design

The current experiments compare CNN, SVM, and Faster R-CNN, but the individual contribution of AlphaPose to the system’s performance remains unquantified. To rigorously evaluate its impact, an ablation study should be conducted by testing Faster R-CNN alone (without AlphaPose) on teacher behavior detection. The results should be added to Table 3 under a new column labeled "Faster R-CNN (w/o AlphaPose)" to demonstrate the performance difference

Validity of the findings

The real-time usability of AlphaPose may be impacted by its high processing complexity. It is recommended that the Experiment section contain an inference speed comparison (FPS) that compares the runtime of several approaches using identical hardware.

There are currently just 74 videos in the collection, which raises questions regarding its generalizability. Has the approach been tried out in classrooms with different subjects or in different countries, for example? It is advised that the description section include a description of the suggested method's capacity for generalization.

Reviewer 2 ·

Basic reporting

The abstract is somewhat lengthy in describing the methodology but lacks a clear summary of contributions.
Structure the abstract into four clear sections: Background, Methodology, Experimental Results, and Key Contributions. the results section could be rewritten as:
“Experimental results show that the proposed method achieves an accuracy of 74.89% in detecting inappropriate behavior, reducing educators’ manual recording time by 47% and decreasing the occurrence of inappropriate behaviors by 63%.”
 Add a concise summary of contributions, such as:
“This study proposes a novel framework integrating AlphaPose and Faster R-CNN for automatic classroom behavior detection, demonstrating the feasibility of computer vision in educational settings.”
• Some sections lack smooth transitions, making it difficult for readers to follow the argument. For instance, in the Introduction, the discussion about computer vision does not transition well into the problem of teacher behavior analysis.
Add transitional sentences to bridge these ideas. For example, after introducing computer vision, include:
“Given these advancements, applying computer vision techniques to automatically monitor and evaluate teacher behaviors presents a promising research direction.”

Experimental design

The manuscript does not mention whether data augmentation techniques (e.g., random rotation, flipping, etc.) were applied. It is recommended to clarify in the Data Processing section whether data augmentation was used and analyze its impact on model performance.
• The experiments in the manuscript are based on the PCB dataset; however, real-world classroom environments may involve different camera angles, lighting conditions, and occlusions, which could affect the model's robustness. It is suggested to conduct additional robustness tests to evaluate the model’s performance under varying conditions.

Validity of the findings

The explanations of figures and tables lack interpretation of the results, merely listing numerical values without discussing their implications.
Add insights and implications after numerical comparisons. For example, in Table 2 (Model Performance Comparison), instead of just listing accuracy values, include an analysis such as:
“As shown in Table 2, the proposed method achieves an F2-score of 0.9309, outperforming CNN (0.8775) and SVM (0.8978). This improvement highlights the advantage of integrating pose estimation with object detection for behavioral classification.”
Each figure and table should conclude with a brief summary explaining the key takeaway.

---

## Round 0.2 · accepted · Accept

Your manuscript has been reviewed by the experts and I'm happy to inform you that your manuscript is judged scientifically sound to be published.

Congratulations and best wishes

Reviewer 1 ·

Basic reporting

I have carefully reviewed the revised version of the manuscript and the authors' responses to my previous comments. I appreciate that the authors have addressed all the concerns and suggestions raised during the initial review, and they have done so thoroughly and effectively. The revisions have significantly improved the clarity, accuracy, and overall quality of the paper.

I congratulate the authors on their excellent work and consider the manuscript now to be in a publishable form. No further revisions are required from my perspective.

Experimental design

.

Validity of the findings

.

Reviewer 2 ·

Basic reporting

The revised manuscript incorporated my concerns with sufficient details and to the extend that satisfy my opinion. The article is now in good shape with necessary details, in the methodology, experimental design and results sections that benefits the research community in this domain.

Experimental design

Revision is accepted

Validity of the findings

Revision is accepted

Additional comments

Paper may be considered for publication now.